# Social Network Addiction and Its Impact on Anxiety Level among University Students

**Raquel Lozano Blasco** 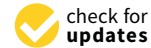**, Cecilia Latorre Cosculluela and Alberto Quílez Robres ***

Department of Science Education, University of Zaragoza, 50001 Zaragoza, Spain; rlozano@unizar.es (R.L.B.); clatorre@unizar.es (C.L.C.)
* Correspondence: aquilez@unizar.es; Tel.: +34-616-484-529

**Abstract:** Despite the obvious favorable effects of social networking sites, there is a risk of developing behavioral addictions. This study aims to analyze addiction to social networks and its relationship with anxiety. A sample (n = 361) of university students (undergraduate, master's and doctoral) comprising 87.5% women and 12.5% men with a mean age of 32.58 (SD = 12.03) and 32.36 (SD = 10.21), respectively, was included. Addiction to social networks was measured using the Social Network Addiction (SNA) questionnaire and anxiety was measured using Spanish adaptation of the Beck Anxiety Inventory (BAI Test). The regression results show how concurrent moderating variables such as age (adulthood) predispose individuals to addiction in some way (Model 5, explained 13.5%, $R^2 = 0.135$, $p = 0.040$). Similarly, we found that the aspect of addiction that generates anxiety is an obsession with social networks. Anxiety arises as excessive use decreases, similar to abstinence syndrome. It is concluded that the harmlessness of social networks and their inappropriate use can lead to behavioral addiction.

**Keywords:** anxiety; internet addiction; social networking sites

## 1. Introduction

The use of the internet has increased in recent years and is becoming a fundamental portal for communication and human interaction that is accessible from multiple devices. Smartphone use is estimated to reach 6.8 billion people in 2022 [1], thus occupying a privileged position as a means of accessing the digital cloud [2–4]. Social networking sites such as Facebook, Twitter and Instagram have recently experienced high growth [5]. Specifically, the author refers to data on the growth of Facebook [6] followers: "as of December 2018, there were around 1.52 billion daily active users on Facebook (Fb) and 2.32 billion active users on the site per month" [5] (p.226). They constitute very favorable spaces in which to solve all kinds of issues, from academic (group learning, acquisition of ICT competences) to affective (communicative skills and social relations). In sociological terms, we find ourselves in a liquid modernity [7,8] which implies that the stage of the big brother to whom we listen and whom we imitate has ended.

Currently, social networks offer multiple examples to follow in the form of figures such as influencers [9]. This means, an influencer is a very active users in social networks whose behaviour is imitated by others. In the words of Bauman and Leoncini [8], an individual chooses which examples to follow and, in this way, assumes the inevitable consequences of doing so. Following this, Ahuja and Alavi [10] state how the internet is consulted as if it were a friend. Social network interactions have therefore come to be regarded as routine relationship practices [11,12]. Their widespread and generalized use raises a question about the psychological needs related to Facebook, Instagram and Twitter use. The Facebook double factor model of Nadkarni and Hofmann [13] stipulates two basic social needs: first, the need to belong to a social group, to be accepted by and integrated into a

community, and second, the need for self-representation, which is concretized by the generation of a concrete idea in third parties. As a result of these needs, it is possible to generate profiles that express one's desired self-representation [14] and permit social experiences [10].

It is worth mentioning that social networking sites also seem to present a certain duality. On the one hand, they can protect people from the appearance of psychological disorders as they allow them to connect, interact and express emotions [15]. It has been clear in previous studies how these platforms have a considerable impact on the quality and quantity of an individual's social interactions; concretely, extraversion was significantly positively correlated with communicative actions: Chat, Messages, Comments, and the Wall [16]. However, the complexity of these relationships has far-reaching implications, and what *a priori* seemed innocuous can lead to serious problems. Undoubtedly, sharing, exhibiting and experiencing are actions that enrich the human being, but excessive exposure to the sweetened lives of other people in social networking sites [14] can cause users to feel dissatisfied with their own realities [10]. Such as, active social contributions correlate with neuroticism, social loneliness and exhibitionism [16].

Over time, they may use social networks more, and their feelings of disappointment may increase to the point of even becoming addicted to these networks. Excessive use and time play a determining role not only in addiction to social networks [17] but also in the emergence of psychological disorders such as anxiety [18]. In fact, addiction to networks is comorbid with other pathologies. Facebook addiction disorder (FAD) is positively correlated with anxiety, depression, insomnia and stress, among other pathologies [17], and is negatively correlated with resilience and physical exercise [19] see Table 1. Alt and Boniel–Nissim [20] explain how the use of the internet is not a problematic practice in itself; rather, it is the abandonment of other activities in favor of staying connected that determines whether a person is addicted [21]. More specifically, it could be argued that when internet use interferes with the individual's every day and daily activities [22], it can cause psychological, physical and social problems [5,18,20,23–25]. In sum, the use of social networks is not intrinsically pathological; rather, it is the obsession with remaining connected at all costs that constitutes a serious problem.

**Table 1.** Correlation between addiction to social networks and anxiety.

|  | Effects | Country | Author |
|---|---|---|---|
| Addiction to social network |  |  |  |
| YouTube | r = 0.32 *** | International | Bérail, Guillon and Bungener [26] |
| Facebook | r = 0.30 *** | USA | Dempesey, O'Brien, Tiamiyu and Elhai [27] |
|  | r = 0.455 ** | Malaysia | Foroughi, Iranmanesh, Nikbin and Hyun [28] |
| Social networks | r = 0.30 *** | USA. | Chen [29] |
| Time or use of social network |  |  |  |
| Facebook | r = 0.33 * | USA | Shaw, Timpano, Tran and Joormann [30] |
|  | r = 0.663 *** | USA | Davidson and Farqhar [31] |
|  | r = 0.07 | Philippines | Labrague [32] |
| Internet addiction |  |  |  |
| Internet | r = 0.411 *** | Israel | Weinstein, Dorani, Elhadif, Bukovza, Yarmulnik, and Dannon [33] |
|  | r = 0.302 ** | China | Hong, Liu, Oei, Zhen, Jiang and Sheng [34] |
|  | r = 0.224 *** | Italy | Casale and Fioravanti [35] |

* $p < 0.05$; ** $p < 0.01$; *** $p < 0.001$

Two decades ago, Young [36–38] and Young, Sandman and Craske [39] exposed the existence of different types of internet addictions: addiction to computer games ("computer addiction"); addiction to surfing the web ("information overload"); addiction to online activities in areas already associated

with addiction, such as online shopping or gambling ("net compulsion"); addiction to cyber-sex ("cybersexual addiction"); and addiction to cyber relationships ("cyber-relationship addiction"). Addiction to social networking sites is encompassed within the latter group [40–42]. In this particular case, it is worth highlighting that social network addiction is a behavioral addiction in which an excessive human–machine interaction is established and which is characterized by basic addiction criteria, such as notoriety, mood modification, tolerance, withdrawal symptoms, conflict and relapse [40,42–44]. This type of addiction has elements in common with other well-known behavioral addictions, such as gambling (online gambling) or gaming (online games) [41,45].

Based on Mamun and Griffiths [42], it is necessary to begin to develop intervention strategies in response to evidence of the problematic use of social networks. Recent research studies have estimated a prevalence of 4% in adolescents [46], while in the young adult population (mean age = 20.72), the prevalence is 39.7% [42]. Regarding the sociodemographic variables of gender and age, the data are inconclusive as there is no consensus among existing studies [47]. However, some studies show that females are more likely to be addicted to social networks, a pattern that they share with younger people [29,48,49]. That is, young adult women comprise a group with a greater likelihood of being addicted to social networks see Table 2.

**Table 2.** Relationship between the sociodemographic variables of gender and age and addiction to social networks.

| | Effects | Country | Author |
|---|---|---|---|
| Gender | | | |
| Facebook | F = 0.30 *** (women) | Norway | Andreassen [48] |
| | F = 21.56 *** (women) | Spain | García–Domingo, Aranda and Fuentes [49] |
| | F = 2.797 * (men) | International | Blanchio, Przepiorka, Benvenuti, Cannata, Giobanu, Senol–Durak and Ben–Ezra [50] |
| YouTube | β = −0.02 | International | Bérail, Guillon and Bungener [26] |
| Networks | Chi$^2$ = 0, df = 1 (not indicated) | India | Mamun and Griffiths [42] |
| | F = 0.30 * (women) | USA | Chen [29] |
| Internet | β = 0.718 | Israel | Weinstein, Dorani, Elhadif, Bukovza, Yarmulnik and Dannon [33] |
| | r = 0.101 ** (not indicated) | China | Hong, Liu, Oei, Zhen, Jiang and Sheng [34] |
| Age | | | |
| Facebook | β = −0.207 *** (young people) | Norway | Andreassen [48] |
| | β = 0.14 ** (elderly adults) | USA | Dempsey, O'Brien, Tiamiyu and Elhai [27] |
| | F = 17.78 *** (young people) | Spain | García–Domingo, Aranda and Fuentes [49] |
| YouTube | β = −0.01; | International | Bérail, Guillon and Bungener [26] |
| Internet | r = 0.205 ** (elderly adults) | China | Hong, Liu, Oei, Zhen, Jiang and Sheng [34] |
| | β = −0.19 *** (young people) | Poland | Błachnio and Przepiork [51] |

* $p < 0.05$; ** $p < 0.01$; *** $p < 0.001$

Considering what has been discussed so far, a problem arises regarding addiction to social networks in a sample of university students (made up of young and middle-aged adults) and its relationship with anxiety disorder. In addition, we intend to analyze the possible effects that other sociodemographic variables, such as gender and age, have on the aforementioned addiction and

anxiety. In line with the data presented in Tables 1 and 2, it is necessary to link the two approaches. Consequently, the study aims to analyze the relationship between addiction to social networks and anxiety in university students, taking into account other variables such as gender and age.

## 2. Materials and Methods

### 2.1. Participants

The sample consisted of 361 undergraduate (2nd, 3rd and 4th), master's and doctoral students at different Spanish universities (Universidad de Zaragoza, Universidad de Castilla–La Mancha) both face-to-face education, online education (Universitat Oberta de Catalunya and Universidad Inernacional de la Rioja) and semi-presential education (Universidad Internacional de Valencia). The selection was the result of a non-probability convenience and snowball sampling in social networking sites of universities. The contact with the participants was made in two phases. Firstly, different groups from all the universities described were contacted through trusted social networks, requesting their anonymous collaboration. Once the desire to participate was expressed, the questionnaire was sent out telematically.

### 2.2. Instruments

First, the Social Network Addiction (SNA) questionnaire of Escurra and Salas [52] was used. With this tool, the variable of addiction to social networks was studied in terms of three factors: a) obsession with social networks; b) lack of personal control over the use of social networks; and c) excessive use of social networks. Obsession with social networks (F1) refers to the individual's mental commitment (recurring thoughts, anxiety, fantasizing and worry about lack of access). Lack of personal control over the use of social networks (F2) relates to the neglect of other tasks and activities in favor of being connected. Excessive use of social networks (F3) encompasses difficulties with controlling use, including the time invested in social networks and the inability to reduce it. In addition, the test includes a section on the degree of knowledge of "friends on social networks" and another on access; the technical language of these sections was adapted to reflect that usually used by the Spanish population. The scale is composed of 24 items answered using a Likert-type scale with an interval of 5 points, from 0 ("never") to 4 ("always").

Second, the Beck Anxiety Inventory (BAI) was used [53]. It is a self-report instrument consisting of 21 items that was initially created to analyze the severity of anxious symptomatology. For each item on the instrument, the respondent indicates the degree to which that statement has applied to him or her in the past week using a 4-point Likert scale on which 0 means "not at all" and 3 means "seriously; I could not stand it". Each of the items receives a score of 0 to 3 based on the response; the sum of the scores for each issue can vary from 0 to 63 points. The internal consistency and construct validity of this instrument are excellent. Specifically, in the sample used for adaptation, a Cronbach alpha index of 0.91 was obtained.

### 2.3. Procedure and Data Analysis

The phenomena and variables were analyzed in their natural context, with no intervention or manipulation. In the first instance, a document of collaboration and consent was sent to the student groups of the different universities through Facebook. The participants, in turn, sent the request for collaboration to other colleagues from their trusted social networks; carrying out a sampling through the snowball method. Upon receipt of this consent, the corresponding questionnaire was sent in digital format via e-mail to each of the participants. This process was carried out in May 2019. Both the procedure and the subsequent data processing were guided by the ethical principles established in the Declaration of Helsinki of the World Medical Association (WMA). Data analysis was performed using IBM SPSS Statistics Visor 24. Initially, descriptive analysis of demographic dates and the results obtained for each factor of the SNA and the BAI test were applied. Then, an ANOVA test was realized,

to determine differences in social network addiction (Factor 1, obsessed with SNs; Factor 2, lack of control regarding SNs; Factor 3, excessive use of SNs) and anxiety with respect to age and sex. Consecutively, a Pearson correlation table was made. Finally, multiple linear regression analysis allowed us to study the influence of some variables (age, sex and anxiety) on social networking sites addiction, thus responding to the ultimate purpose of the study. The data entry methods "forward" was used.

## 3. Results

Demographically Table 3, in terms of gender distribution, 87.5% were women and 12.5% were men. The ages of the participants ranged from 18 to 65 years. Regarding the level of education, 88.9% were enrolled in a bachelor's degree program, 9.1% were enrolled in a master's program and 1.9% were enrolled in a doctoral program. One issue to note is that more participants were enrolled in online universities than in-person programs. These distance-learning universities have a more diverse student profile, and normally, their students are older than those who attend universities in person. For this reason, the average age of the undergraduates was higher than the average age of the students in master's and doctorate programs.

**Table 3.** Sociodemographic characteristics of the sample (M = 361).

|  | **N** | **%** | **Average** | **SD** |
|---|---|---|---|---|
| Gender |  |  |  |  |
| Male | 45 | 12.5 | 32.58 | 12.03 |
| Female | 316 | 87.5 | 32.36 | 10.21 |
| Level of University Study |  |  |  |  |
| Undergraduate | 321 | 88.9 | 32.57 | 10.60 |
| Master's | 33 | 9.1 | 31.64 | 9.55 |
| Doctorate | 7 | 1.9 | 27.29 | 5.22 |
| Age |  |  |  |  |
| 18 to 25 years | 121 | 33.5 | 21.76 | 2.02 |
| 26 to 34 years | 100 | 27.7 | 29.54 | 2.68 |
| 35 to 65 years | 140 | 38.8 | 43.60 | 6.54 |

SD, StandardDeviation

The direct scores obtained for the SNA and BAI psychometric tests see Table 4 showed, in general terms, significant differences between the means of addiction and anxiety according to gender, age and level of university study. Regarding gender, women reported higher scores for the factor "excessive use of social networks ($\mu$ Male = 14.73, SD Male = 5.98, $\mu$ Female = 17.16, SD Female = 6.61, F = −2.33, $p < 0.01$) a $\mu$M = 14.73, DTM = 5.98, $\mu$F = 17.16, DTF = 6.61, F = −2.33, $p < 0.01$) and higher general anxiety score compared to men ($\mu$ Male = 43.53, SD Male = 15.70, $\mu$ Female = 48.04, SD Female = 16.60, F = −1.71, p < 0.05). ANOVAs for the analysis of mean differences as a function of age also yielded significant results. Specifically, for the three factors that comprise addiction to social networks (F1: $F = 13.410$, $p < 0.001$; F2: $F = 19.99$, $p < 0.001$; F3: $F = 8.99$, $p < 0.001$) and anxiety ($F = 12.98$, $p < 0.001$), the youngest participants (between 18 and 25 years) reported higher scores compared with the other two age groups.

**Table 4.** Comparison of mean levels of addiction to social networks and anxiety based on gender, age and level of university study.

| | Factor 1 Obsessed with SNs | | | Factor 2 Lack of Control Regarding SNs | | | Factor 3 Excessive use of SNs | | | Total Addiction | | | Anxiety | | |
|---|---|---|---|---|---|---|---|---|---|---|---|---|---|---|---|
| | M | SD | t/F | M | SD | t/F | M | SD | t/F | M | SD | t/F | M | SD | t/F |
| Gender | | | | | | | | | | | | | | | |
| Male | 53.20 | 19.53 | −1.49 | 12.67 | 5.46 | −1.51 | 14.73 | 5.98 | −2.33 ** | 43.53 | 15.70 | −1.71 * | 11.80 | 1.77 | −1.96 * |
| Female | 58.06 | 20.60 | | 14.06 | 5.85 | | 17.16 | 6.61 | | 48.04 | 16.69 | | 15.34 | 0.63 | |
| Age | | | | | | | | | | | | | | | |
| 18 to 25 years | 64.69 | 21.17 | 13.41*** | 16.35 | 6.09 | 19.99 *** | 18.74 | 6.54 | 8.99 *** | 53.01 | 16.92 | 11.90 *** | 18.72 | 12.22 | 12.98 *** |
| 26 to 34 years | 56.27 | 18.31 | | 13.50 | 5.28 | | 16.68 | 5.06 | | 46.60 | 14.96 | | 14.69 | 10.89 | |
| 35 to 65 years | 52.05 | 19.70 | | 12.04 | 5.19 | | 15.36 | 6.61 | | 43.32 | 16.24 | | 11.76 | 9.97 | |
| Level of university study | | | | | | | | | | | | | | | |
| Undergraduate | 57.48 | 20.34 | 1.20 | 13.91 | 5.80 | 1.22 | 16.92 | 6.49 | 0.616 | 47.50 | 16.45 | 1.14 | 14.77 | 11.21 | 0.424 |
| Master's | 54.94 | 21.32 | | 13.10 | 5.76 | | 15.88 | 7.45 | | 45.42 | 17.66 | | 15.45 | 13.20 | |
| Doctorate | 68.14 | 24.20 | | 16.86 | 6.79 | | 18.57 | 6.75 | | 55.86 | 18.88 | | 18.57 | 11.10 | |

* $p < 0.05$; ** $p < 0.01$; *** $p < 0.001$.

The relationships among the different study variables were subsequently studied see Table 5. There was a significant positive correlation between addiction to social networks and anxiety ($r = 0.232$, $p < 0.001$) and between each of the factors that make up addiction and anxiety (F1: $r = 0.237$, $p < 0.001$; F2: $r = 0.230$, $p < 0.001$; F3: $r = 0.201$, $p < 0.001$); in all cases, it was a low–moderate correlation (Cohen, 2013). On the other hand, there was a significant and moderate negative relationship [25] between age and addiction to social networks ($r = -0.318$, $p < 0.001$). This indicates that, the older the age, the lower the addiction score was (as shown in Table 4). The gender variable was not correlated with total addiction to social networks except for Factor 3 (excessive use), for which there was a low–positive significant relationship (F3: $r = 0.118$, $p = 0.025$).

Considering the significance of the correlations between the personal variables analyzed and addiction to social networks and anxiety, a multiple regression analysis was performed to determine the predictive value of the independent variables gender and age Table 6. Age explained a very significant percentage of anxiety ($R^2 = 7.6\%$, $\beta = -0.276$, $p < 0.000$) as did addiction to social networks (both the total score and each factor) ($R^2 = 8\%$, $\beta = -0.283$, $p < 0.000$). Regarding gender, and although it certainly moderates the relationship between social network addiction and anxiety, it does so with less significance and only for the social network addiction factor of excessive use (F3) ($R^2 = 1.5\%$, $\beta = 0.122$, $p = 0.02$) and for anxiety ($R^2 = 1.1\%$, $\beta = 0.103$, $p = 0.05$). These facts agree with the averages for the direct scores of the psychometric tests discussed above (see Table 3). The contrasts between means allow the observation that the highest scores for addiction and anxiety were obtained by the young adults (18–25 years), a finding that is consistent with the negative value of the Pearson correlations. Additionally, there were differences in terms of gender, with women presenting the highest rates of excessive use (F3) and anxiety.

The effect of addiction to social networks on anxiety levels was also studied using a multiple regression analysis see Table 7. Given that age seemed to play a key role, it was decided to add it along with gender to determine the combined effect of both variables. A total of five multiple regression models were defined:

- Model 1, in which only Factor 1 of addiction to social networks was introduced, explained 6.6% of the total variance ($R^2 = 0.066$, $p = 0.000$).
- Model 2 explained 6.6% ($R^2 = 0.066$, $p = 0.602$),
- Model 3, which included all three factors of addiction to social networks, explained 7.8% of the variance ($R^2 = 0.078$, $p = 0.035$).
- Model 4, in which the three factors of addiction were added along with the age variable, explained 12.5% ($R^2 = 0.125$, $p = 0.000$), and finally,
- Model 5, to which gender was added, explained 13.5% ($R^2 = 0.135$, $p = 0.040$).

**Table 5.** Correlations between scores for addiction to social networks, anxiety, age and gender.

|  | Total Addiction | F 1 Obsessed with SNs | F 2 Lack of Control over SN Use | F 3 Excessive Use of SNs | Anxiety | Age | Gender |
|---|---|---|---|---|---|---|---|
| Total Addiction | - | - | - | - | - | - | - |
| F1. Obsessed with SNs | 0.994 *** | - | - | - | - | - | - |
| F2. Lack of control over SN use | 0.935 *** | 0.945 *** | - | - | - | - | - |
| F3. Excessive use of SNs | 0.950 *** | 0.934 *** | 0.845 *** | - | - | - | - |
| Anxiety | 0.232 *** | 0.237 *** | 0.230 *** | 0.201 *** | - | - | - |
| Age | −0.318 *** | −0.337 *** | −0.377 *** | −0.268 *** | −0.292 *** | - | - |
| Gender | 0.90 | 0.082 | 0.085 | 0.118 * | 0.128 ** | 0.01 | - |

*** $p < 0.001$ ** $p < 0.025$ * $p < 0.05$.

**Table 6.** Standardized and unstandardized regression coefficients for the regression analysis on addiction and anxiety.

| Independent | F 1 Obsessed with SNs | | | | F 2 Lack of Control over SN use | | | | F 3 Excessive Use of SNs | | | | Total Addiction | | | | Anxiety | | | |
|---|---|---|---|---|---|---|---|---|---|---|---|---|---|---|---|---|---|---|---|---|
| | β | *p* | S.E. | b | β | *p* | S.E. | b | β | *p* | S.E. | b | β | *p* | S.E. | b | β | *p* | S.E. | b |
| Gender | 0.078 | 0.137 | 3.26 | 4.86 | 0.079 | 0.132 | 0.92 | 1.39 | 0.122 | 0.020 | 1.04 | 2.42 | 0.090 | 0.089 | 2.64 | 4.50 | 0.103 | 0.050 | 1.80 | 3.54 |
| R² | 0.006 | | | | 0.006 | | | | 0.015 | | | | 0.008 | | | | 0.011 | | | |
| Age | −0.289 | 0.000 | 0.099 | −0.567 | −0.339 | 0.000 | 0.02 | −0.18 | −0.248 | 0.000 | 0.03 | −0.15 | −0.276 | 0.000 | 0.08 | −0.43 | −0.283 | 0.000 | 0.05 | −0.30 |
| R² | 0.083 | | | | 0.115 | | | | 0.062 | | | | 0.076 | | | | 0.080 | | | |

β = Regression coefficient; *p* = Probability; S.E. = Standard Error; b = standardized beta coefficients

**Table 7.** Multiple linear regression analysis for the prediction of anxiety based on age, gender and addiction to social networks.

|  | B | ET | Beta | t | *p*-Value |
|---|---|---|---|---|---|
| MODEL 1 |  |  |  |  |  |
| F1. Obsessed with SNs | 0.142 | 0.028 | 0.256 | 5.022 | 0.000 *** |
| MODEL 2 |  |  |  |  |  |
| F1. Obsessed with SNs | 0.101 | 0.085 | 0.181 | 1.188 | 0.236 |
| F2. Lack of control of SN use | 0.156 | 0.299 | 0.080 | 0.521 | 0.602 |
| MODEL 3 |  |  |  |  |  |
| F1. Obsessed with SNs | 0.266 | 0.115 | 0.479 | 2.314 | 0.021 * |
| F2. Lack of control of SN use | 0.048 | 0.301 | 0.024 | 0.158 | 0.875 |
| F3. Excessive use of SNs | −0.466 | 0.220 | −0.269 | −2.116 | 0.035 * |
| MODEL 4 |  |  |  |  |  |
| F1. Obsessed with SNs | 0.299 | 0.112 | 0.540 | 2.665 | 0.008 * |
| F2. Lack of control of SN use | −0.222 | 0.300 | −0.113 | −0.739 | 0.461 |
| F3. Excessive use of SNs | −0.461 | 0.215 | −0.266 | −2.147 | 0.032 * |
| Age | −0.253 | 0.058 | −0.232 | −4.382 | 0.000 * |
| MODEL 5 |  |  |  |  |  |
| F1. Obsessed with SNs | 0.318 | 0.112 | 0.574 | 2.836 | 0.005 ** |
| F2. Lack of control of SN use | −0.248 | 0.299 | −0.127 | −0.828 | 0.408 |
| F3. Excessive use of SNs | −0.518 | 0.216 | −0.300 | −2.405 | 0.017 * |
| Age | −2.55 | 0.057 | −0.234 | −4.442 | 0.000 ** |
| Gender | 3.545 | 1.718 | 0.103 | 2.063 | 0.040 * |

** $p < 0.025$ * $p < 0.05$.

The multiple regression data (Table 7) showed that obsession with social networks (F1) and age were the main variables that exerted an effect on anxiety, followed by overuse of social networks (F3) and gender. Furthermore, the relationship between age and excessive use (F3) was negative, while the relationship between age and the obsession factor (F1) was positive.

In this way, it was observed that age is a fundamental variable when examining addiction to social networks and its effect on anxiety. Similarly, gender only modulates excessive use and anxiety, and women are the group at highest risk. Lack of personal control over the use of social networks (F2) does not have a direct effect on anxiety; however, obsession with social networks (F1) has a positive relationship with anxiety, and excessive use (F3) has a negative relationship (see Figure 1).

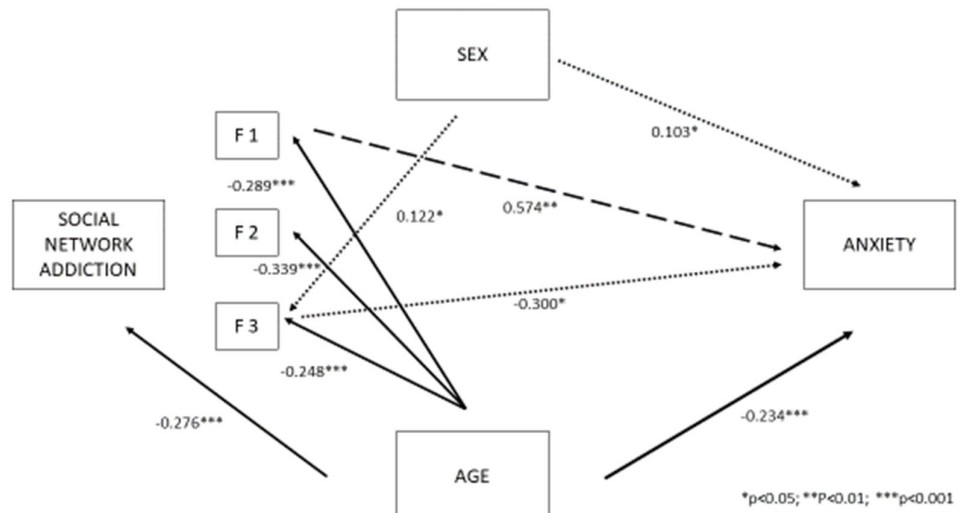

**Figure 1.** Results of the models tested.

## 4. Discussion

The objective of this study was to analyze addiction to social networks and the anxiety derived from it in a sample of university students. In general, the results indicate the existence of an effect between addiction to social networks and anxiety, although its nature varies according to the addiction factor that is studied. The results indicate that there is a moderately significant negative relationship between age and social media addiction. The results are in line with previous studies such as those by Andreassen [48] and García–Domingo, Aranda and Fuentes [49]. Once again, it has been shown that age is a risk factor that makes young adults more vulnerable to both disorders. García–Domingo, Aranda and Fuentes [49] explain how the lack of comparative studies makes it difficult to determine whether the vulnerability of young people is due to an evolutionary stage or greater exposure to social networking sites. In this sense, several authors [54–57] show that young adults participate more actively in social networks than adults of other ages. Koc and Gulyagci [46] point out that easy and fast access to the internet could be a decisive element in understanding addiction and that excessive use of the internet is a triggering factor for addiction [58]. However, recently other authors [27] have stipulated the contrary, since their study found that middle-aged adults (between 35 and 44 years) had higher rates of addiction to social networks compared to young adults (between 26 and 30 years), specifically for networks such as Facebook. It is noteworthy that in the present study, addiction was examined through three factors, which had a different relationship with age. For Factors 1, 2 and 3 (network obsession, lack of control and excessive use), 8.3, 11.5 and 6.2% of the variance, respectively, was explained by age. However, it draws attention to the results on overuse which have an inverse relationship: at younger age and less excessive use, greater anxiety. Furthermore, as healthy use of social networks emerges, anxiety symptoms increase; thus, anxiety appears to be an element of withdrawal syndrome [40,42,43,59]. The symptoms of social networking addiction are similar to those of chemical addiction. They go through withdrawal when they cannot connect. They feel emotional distress (dysphoric mood, insomnia, irritability and psychomotor restlessness). The abuse of social networks correlates with isolation, low academic or work performance, disinterest in daily activities and leisure, behavioral disorders, sedentary lifestyle and obesity. A "snowball effect"is produced. The problems span all areas of life: health, family, school and social relations [60]. In line with previous research [26,27,29,61,62], our results conclude there was a significant positive relationship between addiction to social networks and anxiety. On the other hand, other studies on addiction to YouTube [26] and to the internet in general [34] have not indicated a significant relationship between age and addiction to social networks.

As far as sex is concerned, the correlations did not show a significant relationship between gender and the total score for addiction to social networks. These results are consistent with findings by other authors [26,42,57]. In addition, the multiple regression analysis results indicated that 1.5% of the excessive use of social networks was explained by gender, while obsession and lack of control were not significantly explained by gender. This relationship was already apparent in the correlations, as a slight positive relationship was detected between gender and excessive use. This finding coincides with the results of several authors [34], who studied excessive internet use in relation to gender and found very similar data.

The present study has some limitations that should be taken into account in future research. First, while the number of university students surveyed was relatively high, the various subgroups of students according to gender and level of study were, in some cases, relatively small. It should be noted, however, that this proportionality is attributable to the features that define the population on which the results were generalized. A complementary study could determine the specific study modality (in-person or online) of the students. In addition, it would be interesting to investigate whether there are differences between full-time students and students who combine their academic work with other labor. Furthermore, it is possible that family responsibilities are a protective factor against addiction to social networks. Finally, generalization of the results is limited by the contextualized nature under

which the research was carried out. All these aspects should be assumed in future research while the aforementioned variables are prospectively introduced.

## 5. Conclusions

The relationship of the addiction to social networks with the development of anxiety is verified, highlighting the factors of obsession and excessive use. On the other hand, our study finds that the young population is susceptible to suffer this cormorbidity, placing age as one of the influential variables in this relationship. On the other hand, sex, although presenting a slight significant relationship, does not provide sufficient data for generalization.

Consequently, it is necessary to propose intervention strategies that can be implemented at different educational levels to prevent these situations. In terms of foresight, it would also be useful to study addiction to social networks in greater detail, addressing each subject's individual personality variables as well as their future expectations, locus of control, self-concept and self-esteem and, finally, their personal relationships (with family, friends and spouse). In this way, it would be possible to determine which groups are more likely to experience addiction and to propose intervention strategies in the educational setting at early ages. From this practical perspective, we propose a line of research with longitudinal approaches that allows the study of the improvements obtained after the implementation of programs of healthy social network use for the prevention of state anxiety.

**Author Contributions:** R.L. designed the research study and conducted the search for existing literature. R.L.B. and C.L.C. performed the statistical analysis. A.Q.R. reviewed the theoretical and statistical section making extensions. Finally, the three authors made final revisions of the document before and after translation. All authors have read and agreed to the published version of the manuscript.

**Funding:** This research did not receive any specific grant from funding agencies.

**Acknowledgments:** This research is supported by a contract coverage of the Ministry of Science, Innovation and Universities of Spain (*Formación de Profesorado Universitario* – FPU).

**Conflicts of Interest:** The authors declare that they have no known competing financial interests or personal relationships that could have appeared to influence the work reported in this paper.

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
