# Peer review of "Social Network Addiction and Its Impact on Anxiety Level among University Students"

_sustainability, doi:10.3390/su12135397_

Round 1

Reviewer 1 Report

Dear authors, 

This is a very interesting field of research and I am pleased you are looking into it. I think some clarity is needed within the MS, principally what your methods were and also presentation of the results. I also think you need to make significantly more of the importance of your work - of what it is telling us and what the 'real world' implications of this knowledge are. Specific comments are provided below. I hope these are useful to you. 

Abstract

Don’t need titles within the abstract – look at other articles in the journal for how to set the abstract out. Some sentences seem incomplete (e.g. ‘a sampling of university students’). I would encourage you to include a sentence on why this work is important, why should we care, what does it mean. I would also suggest including some background at the beginning as to why this work is needed. ‘undeniable technological progress’ doesn’t really say much to me in terms of what actually is going to be coming up in the study and why you want to analyse this relationship with anxiety.

References need formatting to be numbered referencing in the main text – reference list should adhere to this

Line 28: I don’t think the phrase ‘social networks’ sounds quite appropriate for these. I think either refer to them as social networking sites/platforms or social media. Same for the use of this term in the abstract. Further along this line is it possible to quantify that growth somehow? I think the fact it is taking over many peoples lives is quite important.

Line 30: what do you mean by issues from academic to affective? Might be beneficial to give examples

Line 34: explain what an influencer is

Line 53/54: not sure how the sentence ‘over time’ really fits in here. I think this paragraph could be built on in terms of linking dissatisfaction with reality to anxiety/depression, maybe insert some numbers if they are available. That starts to provide a reason for your study. I think ‘over time…’ would be better linked to the paragraph below

Line 59: write FAD out in full with (FAD) behind it, then FAD moving forwards

Line 88: you mention nationality in your overall summary but then no mention of nationality in your examples… there’s also no consideration here of the instruments used. What are the different instruments? I’d suggest either adding those details in or removing some of that sentence so you only refer to things you then go on to evidence

Line 99: this has come out of nowhere really – I would suggest maybe adding in some statistics in relation to anxiety in university students, link that to the potential for addiction to social media platforms and then highlight the need for your study. I would then follow this up with ‘the aim of this study is thus to XXX’. I don’t think you need objectives and make sure you are being clear and concise in your aim.

Line 115: should be table 3. How many universities did you target? Was there a reason for those universities/those number of students? What year of undergraduate were they?

‘participants’ paragraph: lots of this is results and should be reported there (all the sociodemographic data – unless you specifically sought out those students). You need instead to identify how students were approached etc so it is of more use if someone were to go and try and repeat this work.

Line 136 onwards: ‘this questionnaire provides evidence’… I don’t think this is needed. You aren’t trying to validate the instrument, you are simply saying you used it. same for L147 – 149.

Line 138 – 140: this is again results – you didn’t have this information before you undertook your study

Procedure and data analysis: this paragraph is very vague and needs more specific details about how the questionnaire was circulated to students (how did you use facebook? Is there a potential bias in your sample population because you targeted students via facebook?) and what analysis was undertaken? How long was the survey up for? What dates?

Line 151 – 153: not sure what these sentences add

Line 162 – 163: details needed here – what went into your models, ‘some variables on others’ is too vague. What variables? How did you analyse the anxiety data?

Line 168: I am not clear what DTM/DTF is?

line 175: is this in relation to anxiety score? If not then I am not clear what is being said here. Needs clarifying

table 4: include a footnote as to what the stars mean. What is ‘total addiction’? how was this computed? It is highly correlated lower down so good to know where this comes from

line 188: that is a very small positive relationship – I would realistically say that yes it is significant but it’s actually relatively meaningless

line 190: this first sentence should be in the methods

line 205: methods

paragraph starting at :205 is very hard to follow – suggest putting more of this information into a table for easier reading

figure 1: needs some explanatory text to say what the numbers/stars refer to

line 229/230: be careful here because your results aren’t necessarily saying that anxiety was caused by the social networks (unless this was a Q you asked?). they could be anxious because they are about to sit exams…

discussion: I’m not sure the discussion concisely reflects what you found and what it means in relation to your aim and in relation to other literature. You have talked about different types of social media platforms in different studies but this didn’t seem to be something you looked into. I also think greater links to what this all means in practical terms is v important. What does it all actually mean? You are v heavy on the gender inequality despite their being <10% difference in mean scores between men and women. Is there the potential issue surrounding lack of confidence in women (research has shown that men will apply for jobs they don’t necessarily think they are capable of because they are confident enough to do so – as an example) and thus a differing use of the scale?

Line 302/303: where did you measure QoL?? If you didn't measure it you need to be careful including statements like this. Unless you want to add in a discussion piece about anxiety and impacts on QoL based on literature. 

Line 312: conclusion shouldn’t have references in – it might be useful to incorporate some of this information into the latter end of your discussion. In fact I would suggest lots of this information goes into the discussion – your conclusion should just be a wrap up/overview of the whole thing. It should draw together all of your main points and provide them with a take home message.

Author Response

Reviewer #1:

Manuscript: sustainability-835336

This letter describes the changes that we have introduced into the main text as a result of the comments of Reviewer 1. We have included the original comments in this document and we have provided an explanation of the changes in the main text as a result of each of them.

We greatly appreciate the first reviewer's comments. We would especially like to thank you for the effort you went to in order to review our research in detail, as well as for your understanding and patience.

Feedback 1. Abstract. Don’t need titles within the abstract – look at other articles in the journal for how to set the abstract out. Some sentences seem incomplete (e.g. ‘a sampling of university students’). I would encourage you to include a sentence on why this work is important, why should we care, what does it mean. I would also suggest including some background at the beginning as to why this work is needed. ‘undeniable technological progress’ doesn’t really say much to me in terms of what actually is going to be coming up in the study and why you want to analyse this relationship with anxiety.

Thank you very much for your contribution. We have restructured the abstract. See lines:10-21.

Despite the obvious favorable effects of social networking sites, there is a risk of developing behavioral addictions. This study aims to analyze addiction to social networks and its relationship with anxiety. A sample (n = 361) of university students (undergraduate, master's and doctoral) comprising 87.5% women and 12.5% men with a mean age of 32.58 (SD = 12.03) and 32.36 (SD = 10.21), respectively, was included. Addiction to social networks was measured using the Social Network Addiction (SNA) questionnaire by Escurra and Salas (2014), and anxiety was measured using Sanz et al.’s (2011) Spanish adaptation of the Beck Anxiety Inventory (BAI Test). The regression results show how concurrent moderating variables such as age (adulthood) predispose individuals to addiction in some way. Similarly, we found that the aspect of addiction that generates anxiety is an obsession with social networks. Anxiety arises as excessive use decreases, similar to abstinence syndrome. It is concluded that the harmlessness of social networks and their inappropriate use can lead to behavioral addiction.

Feedback 2. References need formatting to be numbered referencing in the main text – reference list should adhere to this

References have been formatted and numbered

Feedback 3. Line 28: I don’t think the phrase ‘social networks’ sounds quite appropriate for these. I think either refer to them as social networking sites/platforms or social media. Same for the use of this term in the abstract. Further along this line is it possible to quantify that growth somehow? I think the fact it is taking over many people lives is quite important.

Dear Reviewer, we agree with you on the use of the term "social networking sites".We also consider your other contribution relevant.  We have changed this term throughout the article.  Hemos cambiado este término a lo largo de todo el artículo. We also consider your other contribution relevant. We have added a direct quotation to help you understand the text. See lines:29-33.

Social networking sites such as Facebook, Twitter and Instagram have recently experienced high growth (Badenes, Favris, Gastaldi, Prino, & Longobardi, 2019). Specifically, the author refers to data on the growth of Facebook (2018) followers: “as of December, 2018, there were around 1.52 billion daily active users on Facebook (Fb) and 2.32 billion active users on the site per month” (Badenes, Favris, Gastaldi, Prino, & Longobardi, 2019, p.226).

Feedback 4. Line 30: what do you mean by issues from academic to affective? Might be beneficial to give examples

We have added a few examples in brackets, we hope to improve the understanding of the text. See lines:33-35.

They constitute very favorable spaces in which to solve all kinds of issues, from academic (group learning, acquisition of ICT competences) to affective (communicative skills and social relations).

Feedback 5. Line 34: explain what an influencer is

We have added the following clarification. In other words, an influencer is a very active user in social networks whose sharing is imitated by others. Although, reviewer 2, has exposed us that it was not necessary to make such an extensive reference, because it is something very well-known and current. However, we preferred to clarify the term so that older readers can clearly understand these new linguistic terms. See lines:39-40.

This means, an influencer is a very active user in social networks whose behaviour is imitated by others.

Feedback 6. Line 53/54: not sure how the sentence ‘over time’ really fits in here. I think this paragraph could be built on in terms of linking dissatisfaction with reality to anxiety/depression, maybe insert some numbers if they are available. That starts to provide a reason for your study. I think ‘over time…’ would be better linked to the paragraph below

We agree on the line "Over time, they (...) these network" to the next paragraph. With regard to the other indication, we have added some numerical data from the investigations to reinforce it. See lines:51-63 and 64-65.

It is worth mentioning that social networking sites also seem to present a certain duality.  On the one hand, they can protect people from the appearance of psychological disorders as they allow them to connect, interact and express emotions (Ellison & Boyd, 2013).  It has been clear in previous studies how these platforms have a considerable impact on the quality and quantity of an individual’s social interactions; concretely, extraversion was significantly positively correlated with communicative actions: Chat, r(1158) = .09, p = .003, Messages, r(1158) = .15, p < .001, Comments, r(1158) = .09, p < .001, and the Wall, r(1158) = .13, p < .001 (Ryan & Xenos, 2011). However, the complexity of these relationships has far-reaching implications, and what a priori seemed innocuous can lead to serious problems. Undoubtedly, sharing, exhibiting and experiencing are actions that enrich the human being, but excessive exposure to the sweetened lives of other people in social networking sites (Back et al., 2010) can cause users to feel dissatisfied with their own realities (Ahuja & Alavi, 2017). Such as, active social contributions correlate with neuroticism (r=.08**), social loneliness (r=-.10**) and exhibitionism (r=.09**) (Ryan & Xenos, 2011).

Over time, they may use social networks more, and their feelings of disappointment may increase to the point of even becoming addicted to these networks.

Feedback 7. Line 59: write FAD out in full with (FAD) behind it, then FAD moving forwards

Sorry about the glitch, the change has been executed.  See line: 68.

Facebook addiction disorder (FAD) 

Feedback 8. Line 88: you mention nationality in your overall summary but then no mention of nationality in your examples… there’s also no consideration here of the instruments used. What are the different instruments? I’d suggest either adding those details in or removing some of that sentence so you only refer to things you then go on to evidence

Thank you very much for your observation. We have decided to eliminate this sentence because, as you indicate, it does not provide coherence.

Feedback 9. Line 99: this has come out of nowhere really – I would suggest maybe adding in some statistics in relation to anxiety in university students, link that to the potential for addiction to social media platforms and then highlight the need for your study. I would then follow this up with ‘the aim of this study is thus to XXX’. I don’t think you need objectives and make sure you are being clear and concise in your aim.

We appreciate your contribution, we have simplified and limited the objective of the study. We have also related it to the statistical data previously presented in table 1 and 2. See lines:111-113.

In line with the data presented in Table 1 and 2, it is necessary to link the two approaches. Consequently, the study aims to analyze the relationship between addiction to social networks and anxiety in university students, taking into account other variables such as gender and age.

Feedback 10. Line 115: should be table 3. How many universities did you target? Was there a reason for those universities/those number of students? What year of undergraduate were they?

We appreciate the identification of this typo. We have proceeded to correct it. We have added the universities, two are face-to-face universities and two are online universities. The students were between the second year of undergraduate and graduate studies. See lines: 116-119.

The sample consisted of 361 undergraduate (2nd, 3th and 4th), master's and doctoral students at different Spanish universities (Universidad de Zaragoza, Universidad de Castilla - La Mancha) both face-to-face education, online-education (Universitat Oberta de Catalunya, and Universidad Inernacional de la Rioja, and semi-presential-education (Universidad Internacional de Valencia).

Feedback 11 participants’ paragraph: lots of this is results and should be reported there (all the sociodemographic data – unless you specifically sought out those students). You need instead to identify how students were approached etc so it is of more use if someone were to go and try and repeat this work.

We have extended the information on the type of sampling, to favour the replicability of the study. In addition, we have transferred the demographic data along with table 3 to the results, as suggested. See lines: 119-124 and 165-173.

The selection was the result of a non-probability convenience and snowball sampling in social networking sites of universities. The contact with the participants was made in two phases. Firstly, different groups from all the universities described were contacted through trusted social networks, requesting their anonymous collaboration. Once the desire to participate was expressed, the questionnaire was sent out telematically.

Demographically, in terms of gender distribution, 87.5% were women and 12.5% were men.  The ages of the participants ranged from 18 to 65 years.  Regarding the level of education, 88.9% were enrolled in a bachelor’s degree program, 9.1% were enrolled in a master’s program, and 1.9% were enrolled in a doctoral program. One issue to note is that more participants were enrolled in online universities than in-person programs. These distance-learning universities have a more diverse student profile, and normally, their students are older than those who attend universities in person.  For this reason, the average age of the undergraduates was higher than the average age of the students in master’s and doctorate programs.

Table 3. Sociodemographic characteristics of the sample (M = 361).

Age

N

%

Average

SD

Gender

Male

45

12.5

32.58

12.03

Female

316

87.5

32.36

10.21

Level of university study

Undergraduate

321

88.9

32.57

10.60

Master’s

33

9.1

31.64

9.55

Doctorate

7

1.9

27.29

5.22

Age

18 to 25 years

121

33.5

21.76

2.02

26 to 34 years

100

27.7

29.54

2.68

35 to 65 years

140

38.8

43.60

6.54

Feedback 12 Line 136 onwards: ‘this questionnaire provides evidence’… I don’t think this is needed. You aren’t trying to validate the instrument, you are simply saying you used it. same for L147 – 149.

Thank you very much for your observation, we have proceeded to withdraw it.

Feedback 13 Line 138 – 140: this is again results – you didn’t have this information before you undertook your study. Procedure and data analysis: this paragraph is very vague and needs more specific details about how the questionnaire was circulated to students (how did you use facebook? Is there a potential bias in your sample population because you targeted students via facebook?) and what analysis was undertaken? How long was the survey up for? What dates?

Thank you very much for your contribution. We have added the information you have recommended to us and believe that it greatly enhances the understanding of the study. Regarding the bias of Facebook use, we nuanced the use of a snowball process to extend the scope of population sampling. See lines: 149-153.

In the first instance, a document of collaboration and consent was sent to the student groups of the different universities through Facebook. The participants, in turn, sent the request for collaboration to other colleagues from their trusted social networks; carrying out a sampling through the snowball method. Upon receipt of this consent, the corresponding questionnaire was sent in digital format via e-mail to each of the participants.  This process was carried out last May 2019. 

Feedback 14 Line 151 – 153: not sure what these sentences add

Thank you for your contribution, we have decided to remove this prayer.

Feedback 15 Line 162 – 163: details needed here – what went into your models, ‘some variables on others’ is too vague. What variables? How did you analyse the anxiety data?

Thank you very much for your contribution, we have provided more information. We hope that you will improve the quality and understanding of the study. See lines:156-163.

Initially, descriptive analysis of demographic dates and the results obtained for each factor of the SNA and the BAI Test were applied. Then, an ANOVA test was realized, to determine differences in social network addiction (Factor 1 Obsessed with SNs Factor 2 Lack of control regarding SNs Factor 3 Excessive use of SNs) and anxiety with respect to age and sex. Consecutively, a Pearson correlation table was made. Finally, multiple linear regression analysis allowed us to study the influence of some variables (age, sex and anxiety) on social networking sites addition, thus responding to the ultimate purpose of the study. The data entry methods “forward” was used.

Feedback 16 Line 168: I am not clear what DTM/DTF is?

We apologized for the abbreviation. Instead of DTM, we have put DT Male for male population and DT Female for female population. Same with μ Male and μ Female. Thank you very much for your observation, because it was not clear to the readers. See lines: 178-180.

(μ Male=14.73, DT Male=5.98, μ Female=17.16, DT Female=6.61, F=-2.33, p<.01) and higher general anxiety score compared to men (μ Male=43.53, DT Male=15.70, μ Female=48.04, DT Female=16.60, F=-1.71, p<.05).

Feedback 17 line 175: is this in relation to anxiety score? If not then I am not clear what is being said here. Needs clarifying

Having reviewed this feedback, we have concluded that the information presented is not relevant and leads to confusion. We have decided to delete this sentence.

Feedback 18. table 4: include a footnote as to what the stars mean. What is ‘total addiction’? How was this computed? It is highly correlated lower down so good to know where this comes from

The requested information has been added to the table. Total addiction is the sum of the three factors (F1, F2 and F3). However, it is described in more detail in the section on instruments. See line: 187

*p<.05; **p<.01; ***p<.001

Feedback 19. line 188: that is a very small positive relationship – I would realistically say that yes it is significant but it’s actually relatively meaningless

Thanks for the clarification, we've added it. See lines:

The gender variable was not correlated with total addiction to social networks except for Factor 3 (excessive use), for which there was a low positive significant relationship (F3: r = .118, p = .025).

Feedback 20. line 190: this first sentence should be in the methods

Thank you very much for your contribution. However, in the methods section we have made some changes that reflect this information. We hope it will be of interest to you. See 114-163.

Feedback 21. line 205: methods paragraph starting at :205 is very hard to follow – suggest putting more of this information into a table for easier reading

We have changed the wording to make it clearer and more readable. We hope that we have improved the understanding of the text. See lines:217-224.

  1. Model 1, in which only Factor 1 of addiction to social networks was introduced, explained 6.6% of the total variance (R2 = .066, p = .000).
  2. Model 2, explained 6.6% (R2 = 0.66, p = .602),
  3. Model 3, which included all three factors of addiction to social networks, explained 7.8% of the variance (R2 = .078, p = .035).
  4. Model 4 in which the three factors of addiction were added along with the age variable, explained 12.5% (R2 = .125, p = .000), and finally,
  5. Model 5, to which gender was added, explained 13.5% (R2= .135, p = .040).

Feedback 22. figure 1: needs some explanatory text to say what the numbers/stars refer to

Thank you very much for your indication. We have also added a legend in table 7. See line: 137.

Feedback 23. line 229/230: be careful here because your results aren’t necessarily saying that anxiety was caused by the social networks (unless this was a Q you asked?). they could be anxious because they are about to sit exams…

Thank you for your contribution, we fully agree with your vision. We have chosen to remove that statement as it creates confusion and is too generalistic.

Feedback 24. discussion: I’m not sure the discussion concisely reflects what you found and what it means in relation to your aim and in relation to other literature. You have talked about different types of social media platforms in different studies but this didn’t seem to be something you looked into. I also think greater links to what this all means in practical terms is v important. What does it all actually mean? You are v heavy on the gender inequality despite their being <10% difference in mean scores between men and women. Is there the potential issue surrounding lack of confidence in women (research has shown that men will apply for jobs they don’t necessarily think they are capable of because they are confident enough to do so – as an example) and thus a differing use of the scale?

Thank you very much for your contribution, we have decided to restructure the discussion, organize the findings, and lower the gender discourse. See lines:239-295.

Feedback 25. Line 302/303: where did you measure QoL?? If you didn't measure it you need to be careful including statements like this. Unless you want to add in a discussion piece about anxiety and impacts on QoL based on literature. 

We fully agree with your vision. We have proceeded to remove part of the sentence. We re-write this part. See lines:298-302.

Feedback 26. Line 312: conclusion shouldn’t have references in – it might be useful to incorporate some of this information into the latter end of your discussion. In fact I would suggest lots of this information goes into the discussion – your conclusion should just be a wrap up/overview of the whole thing. It should draw together all of your main points and provide them with a take home message.

Thank you for your clarification, we have implemented a number of changes for your improvement. The limitations have been uploaded for discussion so that this section is "wrap up". We re-write this part. See lines:285-295

Reviewer 2 Report

  1. Introduction

More emphasis should be placed on current social networks and those that will be younger, since it is one of the groups with the greatest impact on social networks and anxiety.

Thus, line 34 makes a very long reference to the issue of influencers and instagram when it is one of the most used social networks currently and by the youngest age groups.

In line 39 and throughout the text, reference is made to Facebook, which is also not one of the most used social networks by younger groups.

A differentiation could be made by age / gender groups, the most used social networks, some characteristics and whether they can be more addictive than others and if the age / gender brackets coincide with those networks that may be more addictive.

Avoid very old references since, due to the current and innovative nature of the subject, the situation has nothing to do with it 10 years ago than now.

Line 73. Two decades ago, Young (1999) exposed the existence of different types of Internet addictions. You should look for more recent data, in addition to this, and if possible from studies that refer to social networks since the Internet is much more than social networks, even the causes of addiction may also be different at least in part.

  1. Materials and methods

113.- Participants

There is a great disparity in the responses regarding gender (Male 45 / female 316), being one of the variables that has been worked to see if there was a relationship between addiction, anxiety and gender and age in the use of social networks, coming to affirm that this relationship exists. You should be more careful with the data and with the results. The conclusions already highlight the limitations and the non-generalization, but more should be explained in this section.

It is a limitation that there are very poorly represented age groups, to qualify the results.

Conclusions

Internet addiction is already experienced by 312 6% of the world population (Cheng and Li, 2014). They should look for more current references and data that focus on the use / addiction of social media.

Author Response

Reviewer #2:

Manuscript: sustainability-835336

This letter describes the changes that we have introduced into the main text as a result of the comments of Reviewer 2. We have included the original comments in this document and we have provided an explanation of the changes in the main text as a result of each of them.

We greatly appreciate the comments of the second reviewer. We have learned to convey information more clearly and concisely. We would like to thank you for your understanding and patience.

Feedback 1. Introduction. More emphasis should be placed on current social networks and those that will be younger, since it is one of the groups with the greatest impact on social networks and anxiety.

We have made a number of changes to the introduction to improve understanding. In the current study we have not taken into account a particular social network. This may be a limitation of the research. We are very grateful for your input as it has given us a new perspective. See lines:24-113.

Feedback 2. Thus, line 34 makes a very long reference to the issue of influencers and instagram when it is one of the most used social networks currently and by the youngest age groups.

Thank you very much for your contribution. We have made a number of changes to clarify the role of influencers to improve understanding as the first reviewer has indicated. We have opted for this vision, so that older readers can have better access to the current language .See lines: 38-40.

Currently, social networks offer multiple examples to follow in the form of figures such as influencers (Yerasani, Appam, Sarma, & Tiwari, 2019). This means, an influencer is a very active user in social networks whose behaviour is imitated by others.

Feedback 3. In line 39 and throughout the text, reference is made to Facebook, which is also not one of the most used social networks by younger groups.

Thank you very much for your contribution. We have made several changes to clarify the role of influencers to improve understanding, as indicated by the first reviewer. We have opted for this view, so that older readers can have better access to the current language. See lines:45-49.

The Facebook double factor model of Nadkarni and Hofmann (2012) stipulates two basic social needs: first, the need to belong to a social group, to be accepted by and integrated into a community, and second, the need for self-representation, which is concretized by the generation of a concrete idea in third parties.

Feedback 4. Avoid very old references since, due to the current and innovative nature of the subject, the situation has nothing to do with it 10 years ago than now.

Thank you for your contribution. We have tried to offer a solution that would respect both your and the reviewer's feedback 1. Consequently, we have kept some older references but accompanied by new studies. See lines: 44.

We have updated the references to liquid modernity according to one of the latest works by Bauman and Leoncini. See lines: 36 and 40.

Other references such as the following have been removed definitively, as their data are similar to more recent research. See lines:

Griffiths, M. (2005). A ‘components’ model of addiction within a biopsychosocial framework. Journal of Substance Use, 10(4), 191-197. doi: 10.1080/14659890500114359

Flisher, C., (2010). Getting plugged in: an overview of internet addiction. Journal of Pediatrics and Child Health, 46, 557-559. doi: 10.1111/j.1440-1754.2010.01879.x

Feedback 5. Line 73. Two decades ago, Young (1999) exposed the existence of different types of Internet addictions. You should look for more recent data, in addition to this, and if possible from studies that refer to social networks since the Internet is much more than social networks, even the causes of addiction may also be different at least in part.

We agree with his vision but Young is the top international reference in Internet addiction and social networks. His research is considered basic. Although, they have been accompanied by more current references from the same author, to give a response more in line with the new reality. See lin:82.

Feedback 6. Materials and methods. 113.- Participants. There is a great disparity in the responses regarding gender (Male 45 / female 316), being one of the variables that has been worked to see if there was a relationship between addiction, anxiety and gender and age in the use of social networks, coming to affirm that this relationship exists. You should be more careful with the data and with the results. The conclusions already highlight the limitations and the non-generalization, but more should be explained in this section. It is a limitation that there are very poorly represented age groups, to qualify the results.

Thank you very much for your clarifications. We have made a number of changes throughout the methodology that we hope will improve methodological clarity and rigour. We have also restructured the results, discussion and conclusions to clearly explain the main findings, emphasizing that sex is a limitation and data cannot be generalized. In this way, we have adapted the response to the contributions of your other review partner. See lines:114-163.

Feedback 7. Conclusions. Internet addiction is already experienced by 312 6% of the world population (Cheng and Li, 2014). They should look for more current references and data that focus on the use / addiction of social media.

Thank you very much for your contribution, we agree. Furthermore, the other reviewer has proposed that we delete this reference. We have therefore removed it from the text.

Reviewer 3 Report

  • I think this is a good paper, very clear and, methodologically substantial,
  • but there is no something new from this article
  • there are so many studies like this, this article only explains the difference sample from the object of research.

Author Response

Reviewer #3:

Manuscript: sustainability-835336

This letter describes the changes that we have introduced into the main text as a result of the comments of Reviewer 3. We thank the reviewer 3 for his kind words and his holistic view of the article. We have made numerous changes in accordance with the other reviewers' instructions. We hope that these have improved the quality of this publication.

In the new version we have tried to enhance the results, modify the introduction and clarify certain confusing aspects of the method and results.

Round 2

Reviewer 1 Report

Dear authors, 

This manuscript is much improved - the methods and results are much clearer and the discussion is more appropriate in terms of your findings. I am glad my earlier comments were useful to you. I have some very minor comments below. 

I have attached a word document with responses to your comments and a couple of extra minor comments. I hope this is useful. 

Author Response

REVIEWER 1

We are grateful for the comments that our article has received from Reviewer 1. We also welcome the opportunity to resubmit the paper in order to take her/his comments into account. We find the comments of Reviewer 1 very constructive

Feedback 1. R: Abstract is much improved. References not needed in the abstract. Line 15 add in ‘a’ after using. Not sure what is meant by line 17 ‘the regression results…’ – clarify what your findings were here.

We make several changes. Thank you very much. See lines: 14-17.

Addiction to social networks was measured using a the Social Network Addiction (SNA) questionnaire, and anxiety was measured using Spanish adaptation of the Beck Anxiety Inventory (BAI Test). The results of the regression show how concurrent moderating variables such as age (adulthood) predispose individuals to addiction in some way (Model 5, explained 13.5%, R2= .135, p = .040

Feedback 2. R: The first reference in your article should be #1 and so on… these might need reordered for the ‘final’ version.

Thank you very much for your contribution. We've made it right.

Feedback 3. R: I think some brief clarity makes sense – especially when a quick google search leads to lots of different definitions! You could probably add this more smoothly into the sentence if you preferred, e.g. ‘[…] influencers [59]; very active users in social networks whose behaviour is imitated by others.’ – although I leave this decision entirely up to you.

Thank you, we consider very interesting. See lines 37 and 38.

This means, an influencer is a

Feedback 4. R: Good detail added in here but for ease of reading I’d remove the statistical outputs (readers can refer to the original MS if they wish to) e.g. social interactions; extraversion as significantly positively correlated with communicative actions e.g.. chat, messages… [51]. [and I would recommend the same for below – keep the detail but drop the statistics]

Thank you very much for your contribution. We agree, this will improve the understanding of the text. See lines 50-58

It has been clear in previous studies how these platforms have a considerable impact on the quality and quantity of an individual’s social interactions; concretely, extraversion was significantly positively correlated with communicative actions: Chat, Messages, Comments, and the Wall [51]. However, the complexity of these relationships has far-reaching implications, and what a priori seemed innocuous can lead to serious problems. Undoubtedly, sharing, exhibiting and experiencing are actions that enrich the human being, but excessive exposure to the sweetened lives of other people in social networking sites [6] can cause users to feel dissatisfied with their own realities [1]. Such as, active social contributions correlate with neuroticism, social loneliness and exhibitionism [51].

Feedback 5 R: Line 69: not sure if ‘person’ should be in this sentence?

Thank you very much for your correction. Indeed, we have reviewed it and there is no need for that word. See lines: 69.

More specifically, it could be argued that when Internet use interferes with the individual’s everyday and daily activities.

Feedback 6 R: Line 84: should ‘addition’ be ‘addiction’?

Thank you very much for your correction.  We have corrected the misprint. See line 84.

Feedback 7 R: Line 141: change ‘last’ to ‘in’

Thank you very much for your correction.  We have corrected the misprint. See line 141.

Feedback 8 R: I have just realised that DT is the same as SD… I would be consistent with whichever you use – as you have used SD in your tables I would probably recommend changing to SD in the text. Can also be written as (mean±SD) so… ‘… Male (mean±SD) = 14.73±5.98; female = 17.16±6.61)…’. Although that is entirely your choice…

Thank you very much for your contribution, we prefer use SD. Lines:165-168.

Feedback 9 R: Line 179/180: Cohen reference needs to be numerical

Thank you very much for your correction. See lines 178-179.

Feedback 10 R: Line 228: remove ‘linked’?

Thank you very much for your correction. See line 227.

Feedback 11 R: Line 234: what is meant by ‘to both disease’?

Dear reviewer we have replaced that term with addiction and anxiety. See lines 233 – 234.

Feedback 12 R: Line 247: is the latter half of this sentence supposed to be in Spanish?

Dear reviewer, we are very sorry about this typo. We have proceeded to draft it in English. See lines 246-247.

However, it draws attention to the results on overuse which have an inverse relationship the younger the age and the less excessive the use, the greater the anxiety.

Reviewer 3 Report

Thank you for improving your manuscript. 

Author Response

REVIEWER 3

We are grateful for the comments that our article has received from Reviewer 3. We also welcome the opportunity to resubmit the paper in order to take her/his comments into account.